# Hierarchical Retrieval: The Geometry and a Pretrain-Finetune Recipe

**Chong You**      **Rajesh Jayaram**      **Ananda Theertha Suresh**      **Robin Nittka**
**Felix Yu**      **Sanjiv Kumar**
Google
{cyou,rkjayaram,theertha,rnittka,felixyu,sanjivk}@google.com

## Abstract

Dual encoder (DE) models, where a pair of matching query and document are embedded into similar vector representations, are widely used in information retrieval due to their simplicity and scalability. However, the Euclidean geometry of the embedding space limits the expressive power of DEs, which may compromise their quality. This paper investigates such limitations in the context of *hierarchical retrieval (HR)*, where the document set has a hierarchical structure and the matching documents for a query are all of its ancestors. We first prove that DEs are feasible for HR as long as the embedding dimension is linear in the depth of the hierarchy and logarithmic in the number of documents. Then we study the problem of learning such embeddings in a standard retrieval setup where DEs are trained on samples of matching query and document pairs. Our experiments reveal a *lost-in-the-long-distance* phenomenon, where retrieval accuracy degrades for documents further away in the hierarchy. To address this, we introduce a *pretrain-finetune* recipe that significantly improves long-distance retrieval without sacrificing performance on closer documents. We experiment on a realistic hierarchy from WordNet for retrieving documents at various levels of abstraction, and show that *pretrain-finetune* boosts the recall on long-distance pairs from 19% to 76%. Finally, we demonstrate that our method improves retrieval of relevant products on a shopping queries dataset.

## 1   Introduction

*Information retrieval* [23] is the task of finding the most relevant documents within a large database in response to a user query. **Dual Encoder (DE)** is one of the most popular modeling architectures for information retrieval due to their simplicity and scalability. It functions by encoding each query and each document by a vector representation, e.g., by using a deep neural network [16, 9, 36]. Then, similarity between a query and a document is calculated using their Euclidean distance or inner product, enabling scalable retrieval through approximate nearest neighbor search [19, 17, 10].

This paper considers **Hierarchical Retrieval (HR)**, a particular case of information retrieval where the set of documents is organized into a (hidden) hierarchical structure. To motivate our retrieval task on a hierarchy, consider the keyword targeting problem in online advertising where ad platforms aim to display ads based on the relevance of the associated keywords to a user's query. While many notions of relevance exist, a particularly important case, known as *Phrase Match* [2, 1, 4], defines relevant keywords as those that are semantically more general than the user's query. This definition motivates the modeling of advertiser keywords as a hierarchy, where higher level keywords are semantically more general than the lower level keywords (see Figure 1 Left). Then, Phrase Match may be expressed as the problem of retrieving not only the keyword that matches exactly the meaning of a user query, but also all those at a higher level in the hierarchy. Motivated by Phrase Match, we

39th Conference on Neural Information Processing Systems (NeurIPS 2025).

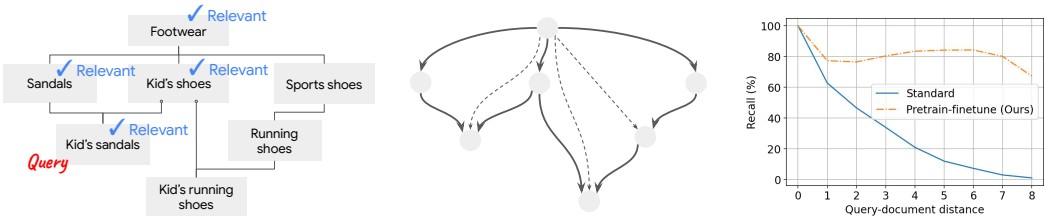

Figure 1: (Left) A illustrative example of a document set that forms a hierarchy. Given a query, e.g. "Kid's sandals", the goal is to retrieve all its ancestors in the hierarchy. (Middle) An abstraction of this hierarchy using a DAG where the edges are represented as solid arrows. For each node $u$ of the graph, the goal of **Hierarchical Retrieval (HR)** is to retrieve all $v$ such that $v$ is reachable from $u$. (Right) The *lost-in-the-long-distance* phenomenon of a Dual Encoders (DEs) trained for HR on the WordNet DAG. Documents further away from the query in the hierarchy are more difficult to retrieve. We introduce a **pretrain-finetune** recipe to drastically improve long-distance retrieval.

define HR as the retrieval problem on a hierarchy that can be described by a direct acyclic graph (DAG, see Figure 1 Middle). Then, the goal is to retrieve, for each node $u$, all nodes $v$ reachable from $u$ with a directed path. In particular, we assume that the DAG is unobserved, which is the typical case in information retrieval where the model is learned on a data set of matching query-document pairs.

**Why is Hierarchical Retrieval hard?** DEs solve information retrieval tasks by finding embeddings such that a relevant document is closer in distance to the query than an irrelevant one. For HR, this distance measure needs to be *asymmetric*. For example, if "Kid's sandals" is the query then "Sandals" is considered a relevant keyword, but the inverse is not true: if "Sandals" is the query then "Kid's sandals" is not a relevant keyword. This makes *asymmetric* DE a natural choice, where the same node is embedded differently by a query encoder $Q()$ and a document encoder $D()$.

However, asymmetric DEs can still be limited for HR due to the properties of the Euclidean geometry [21]. To illustrate this, consider the query "Kid's sandals" for which a DE needs to place the two document embeddings $D($"Sandals"$)$ and $D($"Kid's shoes"$)$ in close proximity to each other, since they need to be both close to $Q($"Kid's sandals"$)$. On the other hand, for the query "Kid's running shoes", a DE needs to place the aforementioned two document embeddings far apart since only one of them is close to $Q($"Kid's running shoes"$)$. This apparent inconsistency leads to the critical question:

> Q1: Does there **exist** Dual Encoders that solve Hierarchical Retrieval?

We will provide an affirmative answer to Q1, establishing the feasibility of DEs for HR.

Nonetheless, the existence of such DEs does not mean that they can be learned from data. In practice, the following question is of great importance:

> Q2: Can we **learn** (from train data) Dual Encoders that solve Hierarchical Retrieval?

We show through experimental evidence that if the embedding dimension of DE is sufficiently high, then the answer to Q2 is positive as well. This result justifies DE as a feasible architecture for HR.

**Lost in the long distance?** While the answer to Q2 is positive with a high embedding dimension, practical retrieval systems have memory and latency requirement hence a low embedding dimension is desirable and critical. Towards improving the practice of DEs for HR, we examine cases where learned embeddings fail due to an insufficient embedding dimension, and discover an intriguing *lost-in-the-long-distance* phenomenon. This phenomenon states that documents further away from the query in the underlying hierarchy are more difficult to retrieve, hence compromises the quality of the retrieval (see Figure 1 Right). To mitigate this, we introduce a *pretrain-finetune* recipe, where a pretrained DE is finetuned on a dataset focusing solely on long-distance pairs. Such a recipe enhances the practicality of DEs for HR by improving long-distance retrieval capabilities.

We summarize the contribution of this paper as follows.

- **Dual Encoders are feasible for Hierarchical Retrieval.** We formally establish that asymmetric DEs are feasible for solving HR. Specifically, with a constructive algorithm that maps an arbitrary DAG to a set of asymmetric embeddings, we prove that such embeddings solve the associated HR task with a high probability. This holds as long as the dimension of the embedding space is larger than a threshold determined by the underlying hierarchy (Section 3).

- **Dual Encoders can be learned from training data to solve Hierarchical Retrieval.** The constructive algorithm above requires the DAG as input. On the other hand, information retrieval tasks often do not directly provide the DAG but require learning embeddings from a training dataset of matching query-document pairs. Next, we conduct an experimental study of this learning problem, starting from a synthetic tree-structured hierarchies for gaining insights. We verify that the learned DEs successfully solve HR with a sufficiently high embedding dimension (Section 4).
- **A Pretrain-Finetune recipe improves the training of Dual Encoders for Hierarchical Retrieval.** We reveal the *lost-in-the-long-distance* phenomenon under a toy setup with a synthetic tree-structured hierarchy. Critically, we show that the standard approach based on rebalancing the sampling of short vs. long distance pairs in the training dataset fails to solve the problem. This highlights the importance of our *pretrain-finetune* recipe, where a pretrained DE further finetuned on long-distance pairs gains enhanced long-distance retrieval capabilities without compromising the quality on short-distance documents (Section 5). Finally, the effectiveness of this recipe extends beyond the toy setup to real datasets, including 1) WordNet, a large lexical database of English, and 2) ESCI, a shopping queries dataset (Section 6).

## 1.1 Related Work

**Graph embedding.** Graph embedding broadly refers to methods that learn a set of node embeddings that preserves certain properties of the graph [35]. While Euclidean embedding is a natural choice, the symmetric nature of the distance metric makes it problematic for handling graphs with directed edges, such as DAGs. To address the issue, numerous works have explored ideas going beyond Euclidean embeddings, e.g., by representing each node with a geometric region [31, 32, 11, 29] or a probability distribution [33, 6]. These ideas may be coupled with non-Euclidean metrics [24, 30, 14] to better model the hierarchical structures. However, such embedding models are not suited for retrieval due to a lack of an efficient nearest neighbor search that hinders their applicability to large scale data. Another work more related to ours is [25], where asymmetric embeddings are used to capture the asymmetry and transitivity of the DAG. However, the focus of [25] is on computing embeddings on a given graph. In contrast, retrieval applications often do not have access to the underlying graph, and embeddings are learned from a dataset of matching query-document pairs.

**Pretrain-finetune.** The paradigm of pretraining and finetuning, where a model is first trained on a large, general-purpose dataset then adapted to downstream tasks with a finetuning procedure, is a cornerstone of modern machine learning [13, 27, 7]. For information retrieval, this paradigm demonstrates effectiveness for improved DE quality, particularly for sparse or noisy downstream data [8]. Our pretraining-finetuning recipe adapts this paradigm by treating the retrieval of long-distance matches as a downstream task, which we demonstrate to be effective in addressing the *loss-in-the-long-distance* challenge in HR.

## 2 Problem Setup

**Hierarchical Retrieval (HR).** Let $\mathcal{Q} = \{q_i\}_{i=1}^n$ be a collection of $n$ queries and $\mathcal{D} = \{x_j\}_{j=1}^m$ be a collection of $m$ documents, respectively. For each $i \in \{1, \dots, n\}$, let $S(q_i) \subseteq \mathcal{D}$ be the set that contains all documents that are the most relevant to the query $q_i$. The goal of information retrieval is to return a ranked list of the documents in $\mathcal{D}$ for each $q_i$, with the top ones being those in $S(q_i)$.

This paper considers HR, a particular case of information retrieval where the document set is associated with a hierarchy. Let us assume that each query has an *exact match* document, e.g., for a query "Sandals for kids", the document "Kid's sandals" from the document set illustrated in Figure 1 is considered an exact match. Then, the relevant document set $S(q_i)$ contains both its *exact match* and all its descendants in the hierarchy. Formally, HR is defined as follows.

**Definition 2.1** (Hierarchical Retrieval (HR)). Assume that there is a directed acyclic graph (DAG), denoted as $\mathcal{G} = (\mathcal{V}, \mathcal{E})$, associated with the document set $\mathcal{D}$, i.e., with $\mathcal{V} = \mathcal{D}$ and $\mathcal{E} \subseteq \mathcal{V} \times \mathcal{V}$. Also, assume that there is a $E : \mathcal{Q} \to \mathcal{D}$, where $E(q_i)$ is referred to as the *exact match* to $q_i$. Then, HR refers to information retrieval with the relevant documents given by $S(q_i) = \{x \in \mathcal{D} : x \text{ is reachable from } E(q_i)\}$ for each $i \in \{1, \dots, n\}$.

**Dual Encoders (DEs).** DEs are embedding models that map the query and document to the same embedding space, where the inner product may be used to measure relevance. A DE is composed of a query encoder, denoted as $f_q(\cdot, \boldsymbol{\theta}_q) : \mathcal{Q} \to \mathbb{R}^d$, and a document encoder, denoted as $f_x(\cdot, \boldsymbol{\theta}_x) : \mathcal{D} \to \mathbb{R}^d$, where $\boldsymbol{\theta}_q$ and $\boldsymbol{\theta}_x$ are parameters to be learned from data.

Following standard practice in information retrieval, we assume that the HR training dataset is composed of matching query-document pairs. That is, there is a collection of pairs $\{(q^{(k)}, x^{(k)})\}_{k=1}^{N} \subseteq \mathcal{Q} \times \mathcal{D}$ that satisfies $x^{(k)} \in S(q^{(k)})$ for each $k = 1, \ldots, N$. The parameters $\boldsymbol{\theta}_q$ and $\boldsymbol{\theta}_x$ of a DE may be learned from minimizing the following softmax loss on the training dataset:

$$L(\boldsymbol{\theta}_q, \boldsymbol{\theta}_x) = \frac{1}{N} \sum_{k=1}^{N} \text{CE}\left(\sigma\left(\frac{\boldsymbol{D}(\boldsymbol{\theta}_x)^\top \cdot f_q(q^{(k)}, \boldsymbol{\theta}_q)}{T}\right), \mathbf{1}_k\right), \tag{1}$$

where

$$\boldsymbol{D}(\boldsymbol{\theta}_x) = \left[f_x(x^{(1)}, \boldsymbol{\theta}_x), \ldots, f_x(x^{(N)}, \boldsymbol{\theta}_x)\right] \in \mathbb{R}^{d \times N} \tag{2}$$

is a matrix containing all document embeddings as columns. In above, $\mathbf{1}_k \in \mathbb{R}^N$ is a vector with the $k$-th entry being 1 and all other entries being 0, and $T$ is a hyper-parameter that is fixed to be 20 in all our experiments. CE means the cross-entropy loss and $\sigma()$ represents the softmax function.

Our ultimate objective is to understand whether a DE from optimizing Equation (1) solves the HR problem. The answer will necessarily depend on multiple factors including the specificity of $\mathcal{G}$, the architecture of $f_q(\cdot, \boldsymbol{\theta}_q)$ and $f_x(\cdot, \boldsymbol{\theta}_x)$, and the optimization algorithm, etc, signficantly complicating the study. In the following, we start with a study of the geometry of HR which is agnostic to the choice of model architecture and optimization procedure.

## 3 The Geometry of Dual Encoders for Hierarchical Retrieval

This section studies the following problem: Is there a collection of query and document embeddings that solves the HR? An affirmative answer to this question asserts the *existence* of embeddings that solve HR, which is a necessary condition for the minimizer of Equation (1) to solve HR.

To answer this question, we start with considering the special case when the embedding dimension $d$ is as large as the size $m$ of the document set. Here, one may simply take $\boldsymbol{x}_j = \mathbf{1}_j$ as the embedding for each $x_j$. Subsequently, we set the query embedding for each $q_j$ as $\boldsymbol{q}_i = \sum_{j \in S(q_i)} \boldsymbol{x}_j$. It can be verified that $\langle \boldsymbol{q}_i, \boldsymbol{x}_j \rangle$ take a value 1 if $j \in S(q_i)$, and 0 otherwise, i.e., these embeddings solve HR.

---

**Algorithm 1** A constructive algorithm for Hierarchical Retrieval

---

1: **Input:** A query set $\mathcal{Q} = \{q_i\}_{i=1}^{n}$, a document set $\mathcal{D} = \{x_j\}_{j=1}^{m}$, and relevant document sets $\{S(q_i) \subseteq \mathcal{D}\}_{i=1}^{n}$.
2: Sample $\{\widehat{\boldsymbol{x}}_j\}_{j=1}^{m} \subseteq \mathbb{R}^d$ *i.i.d.* from the standard Gaussian distribution.
3: For each $j$, take $\boldsymbol{x}_j = \frac{\widehat{\boldsymbol{x}}_j}{\|\widehat{\boldsymbol{x}}_j\|_2}$.
4: For each $i$, take $\boldsymbol{q}_i = \frac{\widehat{\boldsymbol{q}}_i}{\|\widehat{\boldsymbol{q}}_i\|_2}$, where $\widehat{\boldsymbol{q}}_i = \sum_{j \in S(q_i)} \widehat{\boldsymbol{x}}_j$.
5: **Output:** Query embeddings $\{\boldsymbol{q}_i\}_{i=1}^{n}$ and document embeddings $\{\boldsymbol{x}_j\}_{j=1}^{m}$.

---

The construction above is feasible only when the embedding dimension $d$ is allowed to be very large. Towards deriving a tighter bound, we consider another construction where the document embeddings are drawn from a Gaussian distribution, in lieu of the one-hot embeddings, see Algorithm 1. We will use that random embeddings are with a high probability *sufficiently uncorrelated* to each other to show that this construction provides a solution to HR with a much relaxed requirement on $d$. This is stated formally below.

**Theorem 3.1.** *Consider the HR problem in Definition 2.1, and fix any $\epsilon \in (0, 1/2)$. Assume that the hierarchy $\mathcal{G}$ satisfies $|S(q_i)| \leq s, \forall i \in [n]$ for some integer $s$. Then there exists a dimension $d$ with*

$$d = O(\max\{s \log m, 1/\epsilon^2 \log m\}), \tag{3}$$

*a threshold[1] $r$, and a collection of embeddings $\{\boldsymbol{q}_i\}_{i=1}^{n}, \{\boldsymbol{x}_j\}_{j=1}^{m} \subset \mathbb{R}^d$, such that for all $i \in [n]$ and $j \in [m]$ the following holds:*

- *Case 1: If $d_j \in S(q_i)$, then $\langle \boldsymbol{q}_i, \boldsymbol{x}_j \rangle \geq r + \epsilon$.*

- *Case 2: If $d_j \notin S(q_i)$, then $\langle \boldsymbol{q}_i, \boldsymbol{x}_j \rangle \leq r - \epsilon$.*

---

[1]This is a *universal* threshold $r$ that separates the matching documents from no-matching ones for *all* queries. In practice, it is often sufficient to have a query-dependent threshold.

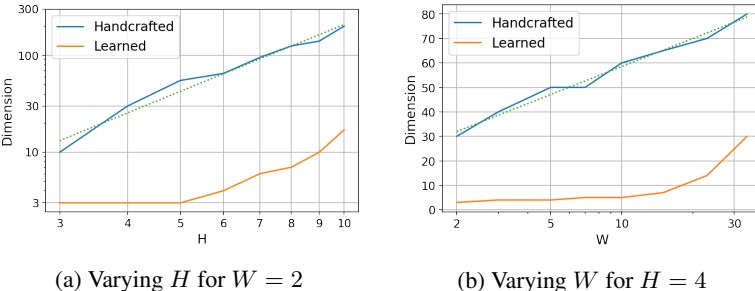

(a) Varying $H$ for $W = 2$         (b) Varying $W$ for $H = 4$

Figure 2: Comparing *learned* (*i.e.*, from optimizing Equation (1)) vs *handcrafted* (*i.e.*, from running Algorithm 1) DEs for a HR task defined on a $W$, $H$-tree. For each $(H, W)$ pair, we experiment with an increasing sequence of embedding dimensions $d$ until the retrieval is successful, and report that dimension on the y-axis. Successful retrieval means that recall is above $95\%$ on the evaluation set. Dotted lines are least squares fittings of the *Handcrafted*.

*Moreover, the vectors $\{\boldsymbol{q}_i\}_{i=1}^n$, $\{\boldsymbol{x}_j\}_{j=1}^m$ can be constructed to satisfy these properties in time $\tilde{O}(md + nsd)$ with high probability.*

Theorem 3.1 asserts that the required embedding dimension is on the order specified in Equation (3). This order is logarithmic in $m$, implying that a small dimension can be sufficient for handling a large document set. For a fixed $\epsilon$ and $m$, this order is linear in $s$, the maximum number of relevant documents per query, which in practice is usually much smaller than $m$. For instance, when $\mathcal{G}$ is a perfect tree with edges pointing from each child node to its parent node, $s$ becomes the number of levels of the tree and is on the order of $\log(m)$; plugging this into Equation (3) we obtain $M = O((\log m)^2)$, which again scales benignly with $m$. Finally, Equation (3) is independent of $n$, the number of queries. This is because the theorem only requires us to handle a subset of queries $\{\boldsymbol{q}_i\}_{i \in T}, T \subseteq \{1, \ldots, n\}$ for which their relevant document sets are distinct from each other, i.e., $S(\boldsymbol{q}_i) \neq S(\boldsymbol{q}_j), \forall \{i, j\} \subseteq T$ and $i \neq j$. For HR, the size of such a $T$ is upper bounded by $m$ according to Definition 2.1.

The construction procedure in Algorithm 1 requires knowing the sets $\{S(q_i)\}_{i=1,\ldots,m}$. In practice, these sets are often not directly observed. Instead, it is often the case that the training dataset is composed of a set of matching query-document pairs, see Section 2. Hence, while Theorem 3.1 establishes the correctness of such embeddings, Algorithm 1 may not be applicable in practice. Instead, the embeddings are often learned by optimizing a proper training loss such as Equation (1). Understanding if such learned embeddings solve the HR problem is the subject of the next section.

**Comparing Theorem 3.1 with Prior Work [15].** Our Theorem 3.1 relates to [15] which also establishes logarithmic bounds on embedding dimensions. However, [15] addresses a different problem. The focus of [15] is multi-label classification, for which their Theorem 2.1 establishes a bound for representing the multi-label set. In contrast, our Theorem 3.1 specifically addresses HR, proving the existence of asymmetric query and document embeddings in Euclidean space that satisfy the specific ancestor-retrieval property. That being said, [15]'s result may be applied to HR by associating their multi-label set with a hierarchy. With this specification, their result states the following (informally):

> There is a dimension $d = O(s \log m)$ such that for $d_j \in S(q_i)$, it has
> $\langle \boldsymbol{q}_i, \boldsymbol{x}_j \rangle > 2/3$, and for $d_j \notin S(q_i)$, it has $\langle \boldsymbol{q}_i, \boldsymbol{x}_j \rangle < 1/3$.

Comparing to this result which has a fixed gap of $1/3$ between matching and no-matching pairs, ours in Theorem 3.1 holds for an arbitrary gap of $2\epsilon$, hence is more general.

## 4 Towards Learning Dual Encoders for Hierarchical Retrieval

This section studies whether learned DEs from optimizing Equation (1) can solve HR. To understand the effect of depth and size of the hierarchy, this section focuses on synthetic, tree-structured hierarchies. Experiments on real hierarchies from practical data are provided in Section 6.

**A toy setup.** We consider *perfect trees*, where each non-leaf node has the same number child nodes and all tree leaf nodes are at the same level. A perfect tree is described by two parameters, namely

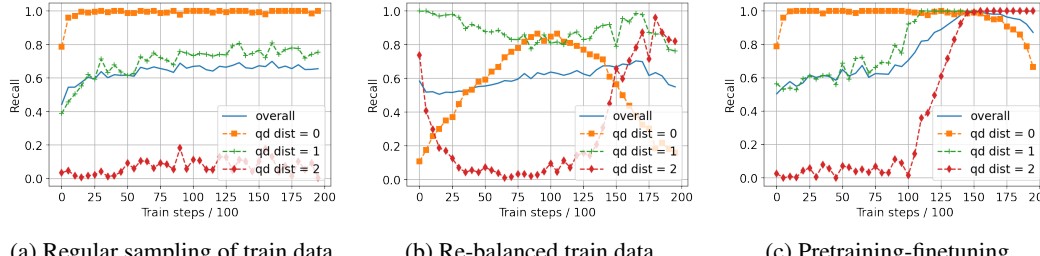

| (a) Regular sampling of train data | (b) Re-balanced train data | (c) Pretraining-finetuning |

Figure 3: Recall on query-document pairs at varying distances $\in \{0, 1, 2\}$ (denoted as *qd dist* in the legend). We train DEs with $d = 3$ dimensional embeddings via optimizing Equation (1) on a tree with $H = 4$ and $W = 5$. (a) Regular training data. The recall for $(q, x)$ pairs with distances 1 and 2 are low. (b) Re-balanced training data, where $(q, x)$ pairs with a distance of 1 or 2 are up-sampled. The recall for such pairs are significantly improved at the cost of a drastic decrease in recall for pairs with distance 0. (c) Pretrain on regular data for 10k steps then finetune on long distance pairs for another 10k steps. Recalls at all distances are close to 100% near 15k steps. In particular, the overall recall (i.e., averaged over the 3 distances) improves to 97%, compared to 66% in (a) and 70% in (b).

the number of child nodes for each non-leaf node (i.e., the *width $W$*), and the number of levels (i.e., the *height $H$*). We use $H, W$-*tree* to refer to a perfect tree with width $W$ and height $H$.

Given an $H, W$-tree, we consider a HR problem where both the query set $\mathcal{Q}$ and the document set $\mathcal{D}$ have a one-to-one correspondence to the set of all nodes of the tree[2]. Then, the hierarchy associated with $\mathcal{D}$ is naturally described by a DAG with edges pointing from each child node to its parent node. Hence, the relevant document set $S(q)$ for a query $q \in \mathcal{Q}$ contains the document that corresponds to the same node in the tree as $q$, as well as all nodes reachable from it. We consider solving this HR problem by a lookup-table DE, a standard choice for studying embedding models [12, 26, 5]. In such a DE, the encoders $f_q$ and $f_x$ are lookup tables with one embedding associated with each $q \in \mathcal{Q}$ and $x \in \mathcal{D}$, respectively.

We sample training data using the following procedure. First, a query $q$ is sampled by drawing a node with equal probabilities from all nodes of the tree. Then, we obtain a matching document to $q$ by sampling a node with equal probabilities from the set of all of its matching documents. Training is conducted by optimizing Equation (1) with gradient descent. Evaluation is conducted on data sampled using the same procedure described above. We use the standard *recall* metric, which is the percentage of $(q, x)$ pairs in the evaluation set for which $x$ is one of the $k$ documents that have the largest inner product score with $q$, with $k = |S(q)|$ being the total number of relevant documents for $q$. We are interested in the smallest embedding dimension (i.e., $d$) sufficient for a successful retrieval. To obtain this dimension, for each $(H, W)$, we experiment with an increasing sequence of $d$ and terminate the process when the evaluation recall metric is >95%.

**Results.** In Figure 2, we report the dimension $d$ needed for a successful retrieval as a function of $(H, W)$. Towards that, we vary $H$ for a fixed $W = 2$ on the left, and vary $W$ for a fixed $H = 4$ on the right. Both cases show that a reasonably large $d$ is sufficient even for $H$ up to 10 and $W$ up to 30.

We further compare such *learned* embeddings with *handcrafted* ones from Algorithm 1. From Theorem 3.1, the handcrafted embeddings solve HR with $d = O(s \log m)$. For $H, W$-trees, we have $s = H$, $m = O(W^{H-1})$, which gives $d = O(H^2 \log W)$. This aligns well with our simulation results which we report in Figure 2. For example, in Figure 2a, we perform a line fitting in the log-log space and obtain a slope of 2.29, whereas the slope derived from $d = O(H^2 \log W)$ is 2. In Figure 2b, we perform a line fitting in the space of $\log(W)$ and obtains a slope of 16.5, whereas the slope derived from $d = O(H^2 \log W)$ is 16. Finally, Figure 2 shows that that learned embeddings achieve successful retrieval with a much smaller $d$ compared to our handcrafted embeddings.

## 5 Improving Dual Encoders for Hierarchical Retrieval

With the establishment that standard training of DEs solves HR, this section takes one step further and asks the following practical question: Can we improve our training algorithm to minimize the dimension required for solving HR? We approach this by examining the failure cases of learned

---

[2]Excluding the root node since if it were included then all queries should trivially retrieve it.

embeddings from standard training *when the dimension is insufficient*. This leads us to discover a common failure case called the *lost-in-the-long-distance* phenomenon. In addressing this issue, we present a pretrain-finetune recipe that leads to an improved retrieval quality.

## 5.1 *Lost-in-the-Long-Distance*

We again consider HR on a $H, W$-tree as described in Section 4, but focus on a particular case where standard training of DE fails to retrieve relevant documents. In particular, we consider the case of $H = 4$, $W = 5$, and $d = 3$. Towards understanding this failure case, we introduce the notion of distance between a matching pair $(q, x)$, defined as the difference between the level of the tree nodes corresponding to $q$ and $x$. For example, a distance of 0 means that $q$ and $x$ correspond to the same node, and a distance of 1 means that $x$ corresponds to the parent node of $q$.

For the tree with $H = 4$, $W = 5$, any query that corresponds to a leaf node has 3 matching documents with distance 0, 1, and 2, respectively (recall that the root node does not correspond to any query / document). We evaluate recall for query-document pairs at these three distances separately, and report results in Figure 3a. It can be seen that the learned embeddings achieve almost perfect retrieval for pairs with a distance 0, but do not work well for pairs with distance 1 and 2. We refer to the phenomenon that matching documents at longer distances to the query tend to be lost in retrieval as *lost-in-the-long-distance*.

**Failure of re-balanced sampling.** A tempting approach to alleviate *lost-in-the-long-distance* is to re-balance the training set, so that more pairs of longer distances are included. To test this, we consider two sampling distributions:

- *Regular* sampling which refers to the sampling procedure described in Section 4. For $H = 4$, $W = 5$, distances 0, 1, and 2 pairs are sampled with probabilities 38%, 35%, and 27%, respectively.
- *Heavy-Tail* sampling, where pairs with distances 0, 1, and 2 are sampled with probabilities 0%, 50%, and 50%, respectively.

By mixing regular and heavy-tail sampling with a ratio of $p : 1 - p$, we may create training datasets with a controllable ratio between short and long distance pairs. In Figure 3b we report the result with $p = 0.03$. It can be seen that the recall for pairs with distance 1 and 2 are significantly improved and reaches a level of beyond 80% towards the end of training. However, this comes at the cost of a significant recall degradation on distance 0 pairs. Finally, this tradeoff cannot be fixed by tuning $p$, as illustrated in Figure 6 (see Appendix) which contains further results with varying $p$ in $\{0.01, 0.1, 0.3\}$.

## 5.2 Main Algorithm: A Pretrain-Finetune Recipe

We introduce a pretrain-finetune recipe to address the challenge of *lost-in-the-long-distance*. This approach simply means that the DE is first pretrained on a standard training set, then finetuned on a long-distance dataset. Notably, the finetuning stage requires long-distance data *only* and does *not* require tuning the ratio of short vs long distance pairs as a hyperparameter.

We conduct an experiment with pretraining and finetuning using data from regular and heavy-tail sampling, respectively, and report the results in Figure 3c. We observe that in the finetuning stage, the retrieval quality for pairs with distance 1 and 2 quickly improves and reaches nearly 100% at 15,000 train step. Notably, at this point the recall for distance 0 pairs remains close to 100%, and the overall recall (i.e., averaged over pairs of all distances) is 97%, far exceeding the regular data sampling (which has 66% recall) or re-balanced data sampling (which has 70% near 17000 steps). Finally, after 15,000 steps the quality of distance 0 pairs starts to decline. This is expected since the finetuning stage does not have any training data with distance 0. However, this quality degradation does not compromise the practicality of our approach since one can apply early stopping during the finetuning stage by monitoring the model quality on a validation set.

**Discussion on data requirement.** In applying the pretrain-finetune recipe, a practical question is how to construct the long-distance dataset for finetuning when the underlying hierarchy, and thus the query-document distances, is unobserved as is typical in many retrieval applications. The key point is that our recipe does not require precise path lengths or knowledge of the full DAG. Instead, it only requires a practical proxy for distance that can be used to partition the training data into short-distance and long-distance subsets. This proxy is often readily available from the data or the problem definition itself. For instance, in our shopping dataset experiment (see Section 6), we treat Exact query-product matches as the short-distance set for pretraining and Substitute matches as

Table 1: Quality of DE for HR on WordNet. *Regular sampling* refers to first sampling a query then a document uniformly at random from the set of all matching documents. *Rebalanced* means a mixture of data from regular sampling with a proportion $p$ and a heavy-tail data of proportion $1 - p$ where long-distance pairs are upsampled proportionally to their distance. *Pretrain-finetune* (ours) refers to first pretraining on regular sampling data then finetuning on heavy-tail data. Quality is measured by averaged recall on a test set. *Our method (i.e., pretrain-finetune) enables good retrieval quality for query-document pairs at all distances.*

| Method | Query-document distance | | | | | | | | | Min | Overall |
|---|---|---|---|---|---|---|---|---|---|---|---|
| | 0 | 1 | 2 | 3 | 4 | 5 | 6 | 7 | 8 | | |
| *Embedding dimension = 16:* | | | | | | | | | | | |
| Regular sampling | **100.0** | **62.8** | **46.6** | 33.9 | 20.9 | 11.9 | 7.2 | 2.9 | 1.0 | 1.0 | 43.0 |
| **Pretrain-finetune (Ours)** | **100.0** | 57.1 | 46.4 | **47.9** | **50.2** | **53.6** | **53.1** | **47.3** | **32.0** | **32.0** | **60.1** |
| *Embedding dimension = 32:* | | | | | | | | | | | |
| Regular sampling | **100.0** | **90.8** | **79.2** | 62.3 | 46.4 | 31.8 | 20.1 | 14.8 | 8.4 | 8.4 | 61.8 |
| **Pretrain-finetune (Ours)** | **100.0** | 77.3 | 76.5 | **80.4** | **83.5** | **84.2** | **84.3** | **80.1** | **67.3** | **67.3** | **87.3** |
| *Embedding dimension = 64:* | | | | | | | | | | | |
| Regular sampling | **100.0** | **93.9** | 86.9 | 76.8 | 60.2 | 46.8 | 36.1 | 28.8 | 19.4 | 19.4 | 71.4 |
| Rebalanced (p=0.01) | 0.6 | 46.9 | 69.7 | 67.2 | 56.3 | 44.7 | 39.4 | 34.9 | 36.6 | 0.6 | 41.8 |
| Rebalanced (p=0.03) | 2.8 | 49.6 | 69.4 | 64.2 | 52.1 | 43.2 | 37.7 | 31.8 | 32.7 | 2.8 | 40.7 |
| **Pretrain-finetune (Ours)** | **100.0** | 90.8 | **91.6** | **92.7** | **92.6** | **91.8** | **90.9** | **87.3** | **75.7** | **75.7** | **92.3** |

the long-distance set for finetuning. In other scenarios, this partition could be based on whether a document is a direct parent versus a more remote ancestor in a known but partial hierarchy. Finally, human annotation can be another viable path towards obtaining such a dataset, which is a significantly easier task than annotating the full DAG. This flexibility allows our pretrain-finetune recipe to be applied in a wide range of practical settings where the full hierarchy is not explicitly given.

Finally, our recipe implicitly assumes that there is sufficient short-distance data to learn a meaningful initial representation during pretraining. If this data is extremely sparse, pretraining may be ineffective, and a mixed training approach might indeed perform better.

# 6 Experiments on Real Data

In this section, we experiment with the pretrain-finetune recipe on two real datasets, namely WordNet and ESCI. On WordNet, which is a large lexical database of English, our method improves the retrieval of hypernyms that are several levels more general than the query. On ESCI, which is a shopping queries dataset where each query has both exact matching products and substitute products, our method enables a single DE to retrieve both categories at a higher recall.

## 6.1 WordNet Experiments

WordNet [22] is a large lexical database of English where the nouns, verbs, adjectives, and adverbs are grouped into *synsets* that represent synonyms. The set of synsets is equipped with a binary *hypernym* relation, e.g., "chair" is the hypernym of "armchair". This relation may be described by a DAG with nodes corresponding to synsets and edges pointing from a synset to its hypernym synset.

In our experiments, we use the 82,115 noun synsets as our document set $\mathcal{D}$. We take the query set $\mathcal{Q}$ to be the same as $\mathcal{D}$. For each query $q \in \mathcal{Q}$, the matching documents $S(q)$ include itself, its hypernyms, and hypernyms of all hypernyms, etc. For example, matching documents for the query "cat" include "cat", "feline", "carnivore", "placental", etc. In practice, we make a slight modification to this definition by restricting to $(q, d)$ pairs with a distance of at most 8. Here, the distance between two synsets is defined as the length of the shortest path that connects them in the hypernym DAG.

Unless specified otherwise, we use the following *regular sampling* procedure to generate training and evaluation data. First, a query $q$ is sampled uniformly at random among all 82,115 synsets. Then, a document is sampled uniformly at random from the set of all matching documents to $q$.

***Lost-in-the-long-distance.*** We train a lookup-table DE by optimizing Equation (1) using SGD for 50k iterations on 10M matching pairs from regular sampling. We use learning rate 0.5, momentum 0.9, and batch size 4096. To evaluate the learned DE, we use the recall metric defined as the percentage of $(q, x)$ pairs for which $x$ is one of the $k$ documents that have the largest inner product score with $q$, with $k = |S(q)|$. We use a validation set of size 10k to pick the best checkpoint. Then,

Table 2: Spearman correlation score $\rho$ on Hyperlex for DE trained on WordNet. We used 5-dimensional embeddings to be consistent with prior work. *Our method (i.e., pretrain-finetune) obtains the best correlation score.*

| **Method** | OrderEmb [34] | WN-Basic [34] | WN-Euclidean [24] | Regular sampling | **Pretrain-finetune (ours)** |
|---|---|---|---|---|---|
| $\rho$ | 0.195 | 0.240 | 0.389 | 0.350 | **0.415** |

we report in Table 1 the recall computed on a test set of size 10k, including an overall recall that is averaged over all pairs in the test set, and recall on slices with different query-document distances (i.e., 0, 1, ..., 8). For a varying dimension of the embedding space in $\{16, 32, 64\}$, we observe that quality degrades rapidly as a function of the distance between the query and document. A similar qualitative behavior is also observed in [18].

**Rebalanced data sampling is insufficient.** The *lost-in-the-long-distance* phenomenon may be attributed to the distribution of data from regular sampling, which is biased towards pairs with short distances (see Figure 4). A natural choice is to use a heavy-tail sampling of the training dataset, which works as follows. First, a query $q$ is sampled uniformly at random from all synsets. Then, the matching document for $q$ is sampled with a probability *proportional to the distance between the document and $q$.*

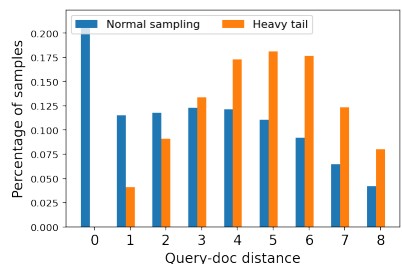

Figure 4: Distribution of regular sampling and heavy-tail data over varying query-document distances on WordNet.

We create a *rebalanced* dataset where each batch has $p \times 4096$ pairs from regular sampling and $(1-p) \times 4096$ from heavy-tailed data. Results with $p = 0.01$ or $p = 0.03$ for embedding dimension 64 are reported in Table 1. It can be seen that rebalanced data improves the retrieval quality on long distance pairs but at the cost of compromising quality on short distance pairs, aligning with the observation in Section 5.

**Our pretrain-finetune recipe offers a solution.** We pretrain the DE on data from normal sampling, then finetune on the heavy-tail dataset. During finetuning, we reduce the learning rate to 1,000 times smaller and increase the temperature in Equation (1) from 20 to 500; an ablation study on these two hyper-parameters is provided in Appendix E. In both of the two stages, we pick the best checkpoint on the validation set. The results in Table 1 demonstrate a significant retrieval quality improvement for long distance document, leading to much higher overall recall. We further provide an example of the retrieved documents for selected queries in Table 3. These examples show that regular sampling tends to miss the long distance pairs and the pretrain-finetune recipe fixes many such errors.

**Hypernymy evaluation.** We supplement our evaluation by using HyperLex [34], a dataset for evaluating how well a model captures the hyponymy-hypernymy relation between concept pairs. Here, we evaluate our DE models learned on WordNet using regular sampling as well as the pretrain-finetune recipe. We also compare with results from previous papers and report the results in Table 2. This result confirms the effectiveness of the pretrain-finetune recipe.

## 6.2   Experiment on ESCI Shopping Dataset

ESCI [28] is a public Amazon search dataset, containing 2.6 million manually labeled query-product relevance judgements in four categories, namely, *Exact, Substitute, Complement,* and *Irrelevant.* For our experiment, we focus on *Exact,* where the product is relevant for the query and satisfies all query specifications, and *Substitute* where the product is somewhat relevant and fails to fulfill some aspects of the query. We consider the task of retrieving, given a user query, both Exact and Substitute documents by a DE[3].

**Training and evaluation data.** ESCI comes with a train vs test data splitting. We take the Exact and Substitute pairs from the train split as our training sets, denoted as $E_{\text{train}}$ and $S_{\text{train}}$, respectively.

---

[3]This task may not fit exactly the HR problem definition. However, the similarity to HR is that a Substitute match may be considered as having a longer distance to the query than an Exact match, leading to the same *lost-in-long-distance* challenge as HR.

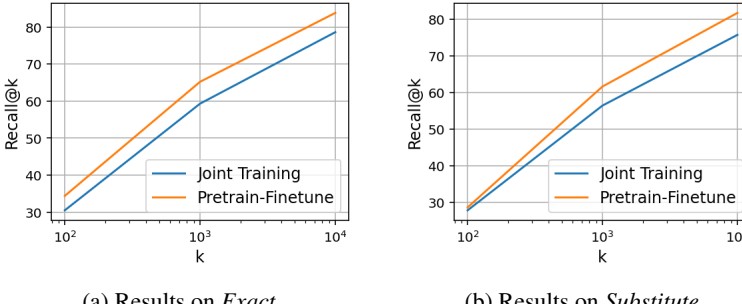

(a) Results on *Exact*          (b) Results on *Substitute*

Figure 5: Quality of *Exact* (left) and *Substitute* (right) retrieval on the ESCI dataset using a single DE. By first pretraining on Exact then finetuning on Substitute matches, *our pretrain-finetune recipe performs better than naively joint training on Exact and Substitute matches.*

$E_{\text{train}}$ and $S_{\text{train}}$ contain 1.3 million and 0.4 million matches, respectively. We sample 5k Exact and 2k Substitute pairs from the test split for evaluating our model. These two sets are denoted as $E_{\text{test}}$ and $S_{\text{test}}$, respectively. To evaluate our model, we use

$$\text{Recall@}k = \frac{\#\{(q,d) \in T \mid d \text{ is among the top-}k \text{ matches for } q\}}{|T|}, \tag{4}$$

where $T$ is either $E_{\text{test}}$ or $S_{\text{test}}$. In words, it is the percentage of $(q, x)$ pairs in the set $T$ with the property that the inner product score between $q$ and $x$ is among the $k$ largest ones across all $x \in \mathcal{D}$. Here, $\mathcal{D}$ is all products provided as part of ESCI and has a size of approximately 1 million.

We use the SentencePiece tokenizer, Transformers for the encoder models in DE, and the Lazy Adam optimizer [3]. Details are provided in Appendix B.

**Methods and Results.** A naive approach for this task is *Joint Training*, where DE is trained on the union of $E_{\text{train}}$ and $S_{\text{train}}$. We compare this with our pretrain-finetune recipe, where a DE is pretrained on $E_{\text{train}}$ then finetuned on $S_{\text{train}}$. The results are presented in Figure 5 on Exact matches (Left) and Substitute matches (Right). It shows that the pretrain-finetune recipe performs better than joint training in terms of recall@k for varying values of $k \in \{100, 1000, 10000\}$.

## 7 Conclusion

This paper studies the theory and practice of dual encoders (DE) for hierarchical retrieval (HR), the task where the document set is organized into a hierarchy. Through a geometric analysis, we first validated rigorously that DEs are capable of solving the HR problem despite the constraints from the Euclidean geometry. We then demonstrated through experiments that such DEs can be found in practice via standard DE training. Towards improving the practical performance of DE, we introduced a pretrain-finetune recipe which addresses the challenge associated with long-distance pairs. Finally, the effectiveness of this recipe is verified on real datasets including WordNet and ESCI shopping queries.

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

# A Proof of Theorem 3.1

We now present the proof that our construction of embeddings solves the dual encoder embedding construction task. In what follows, we write $a = x \pm y$ to indicate the containment $a \in [x - y, x + y]$, and we assume $n = m$ without loss of generality.

*Proof of Theorem 3.1.* We begin by drawing standard Gaussian vectors $x_1, \ldots, x_n \sim \mathcal{N}(0, I_d) \in \mathbb{R}^d$ (we will later normalize them). Next, we set $q_i = \frac{1}{\sqrt{|S_i|}} \sum_{j \in S_i} x_j$ (here we abuse notation and think of $S_i \subset [m]$ to be the corresponding indices). Note that, by stability of Gaussian random variables and independence of the $x_j$'s, it follows that each coordinate of $q_i$ is distributed independently as a standard normal distributed (i.e. $\mathcal{N}(0, 1)$). By standard $\chi^2$ concentration (e.g. Lemma 1 [20]), for any single Gaussian vector $g$ and value $\lambda > 0$, if $g \sim \mathcal{N}(0, I_d)$ is a vector of i.i.d. standard Gaussian variables, we have $|\|g\|_2^2 - d| \leq 2\sqrt{d\lambda} + 2\lambda$ with probability at least $1 - 2 \cdot 2^{-\lambda}$. Setting $\lambda = 4 \log n$ and taking $d = \Omega(\log n)$ with a sufficiently large constant, we have

$$\mathbf{Pr}\left[\left|\|g\|_2^2 - d\right| \leq 5\sqrt{d \log n}\right] \geq 1 - n^{-4}$$

We can thus condition on $\|x_i\|_2^2 = d \pm 5\sqrt{d \log n}$ and $\|q_i\|_2^2 = d \pm 5\sqrt{d \log n}$ occurring for all $i \in [n]$, which holds with probability at least $1 - 2n^{-3}$ by a union bound over the $2n$ vectors. Call this event $\mathcal{E}_1$.

Let $E$ be the edge set of the HR problem, namely, $(i, j) \in E$ iff $j \in S_i$. To further analyze the construction, first define the event $\mathcal{E}_2$ that for all $(i, j) \notin E$, we have $|\langle q_i, x_j \rangle| \leq 100\sqrt{d \log n}$. Also define the event $\mathcal{E}_3$ that for all $(i, j) \in E$ we have $|\langle \sum_{t \in S_i \setminus j} x_t, x_j \rangle| < 100\sqrt{ds \log n}$.

We first analyze $\mathbf{Pr}[\mathcal{E}_2]$. If $(i, j) \notin E$, then $q_i$ and $x_j$ are independent Gaussian vectors, thus by Gaussian stability we have

$$|\langle q_i, x_j \rangle| \sim |g| \cdot \|x_j\|_2 \leq |g|\sqrt{d}\left(1 + 5\sqrt{\frac{\log(n)}{d}}\right)^{1/2} \leq |g|\sqrt{d}(1 + \tfrac{1}{100})$$

where $g \sim \mathcal{N}(0, 1)$ and we took $d = \Omega(\log n)$. Via the density function of a Gaussian, we have $\mathbf{Pr}\left[|g| \cdot \|x_j\|_2 > 100\sqrt{d \log n}\right] < 1/n^4$. Thus, by a union bound over at most $n^2$ pairs, we have $\mathbf{Pr}[\mathcal{E}_2] > 1 - 1/n^2$. For $\mathcal{E}_3$, note that for any $i \in [n]$ with $j \in S_i$, the vector $\sum_{t \in S_i \setminus j} x_t$ is distributed like $\mathcal{N}(0, \sqrt{|S_i| - 1} \cdot I_d)$; namely each coordinate is i.i.d. Gaussian distributed with variance $|S_i| - 1$. Thus

$$\left|\left\langle \sum_{t \in S_i \setminus j} x_t, x_j \right\rangle\right| \sim \sqrt{|S_i| - 1} \cdot |g| \cdot \|x_j\|_2 < |g|\sqrt{sd}(1 + \tfrac{1}{100})$$

where again $g \sim \mathcal{N}(0, 1)$. Following the same argument as above yields $\mathbf{Pr}[\mathcal{E}_3] > 1 - n^{-2}$.

In what follows, let $\gamma = 10 \cdot \max\{s, \frac{1}{\epsilon^2}\}$, and set the dimension $d = C\gamma \log n$ for a sufficiently large constant $C$. Conditioned on $\mathcal{E}_1, \mathcal{E}_2, \mathcal{E}_3$, we claim that the vectors $q_1/\|q_1\|_2, \ldots, q_n/\|q_n\|_2, x_1/\|x_1\|_2, \ldots, x_n/\|x_n\|_2$ satisfy the desired properties with threshold $r = \frac{1}{4\sqrt{\gamma}}$. For case one, if $j \in S_i$ we have

$$\left\langle \frac{q_i}{\|q_i\|_2}, \frac{x_j}{\|x_j\|_2} \right\rangle = \frac{1}{\|q_i\|_2 \|x_j\|_2 \sqrt{|S_i|}} \left(\|x_j\|_2^2 + \left\langle \sum_{t \in S_i \setminus j} x_t, x_j \right\rangle\right)$$

$$\geq \frac{2}{3d\sqrt{\gamma}}\left(\frac{2}{3}d - 100\sqrt{ds \log n}\right) > \frac{4}{9\sqrt{\gamma}} - \frac{200}{3} \cdot \sqrt{\frac{\log n}{d}} \tag{5}$$

$$\geq \frac{1}{3\sqrt{\gamma}}$$

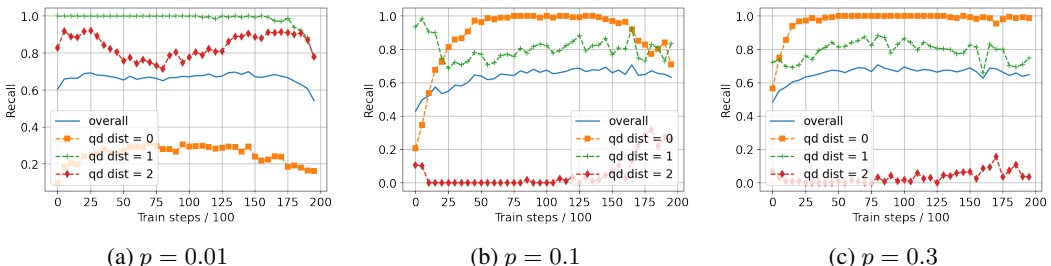

| | | | | | |
|---|---|---|---|---|---|
| (a) $p = 0.01$ | | (b) $p = 0.1$ | | (c) $p = 0.3$ | |

Figure 6: Effect of re-balanced training data with a varying ratio $p$ between the regular and the heavy-tail data on retrieval quality.

Table 3: Retrieved synsets for selected queries with 64-dimensional embeddings. For each of the three queries considered here, we list the relevant documents in the row "Groundtruth" with an ascending order in their distance to the query. For "Regular sampling" and "Pretrain-finetune", we list top-$k$ documents in an ascending order of $k$. Documents retrieved but not in the groundtruth are underscored.

| | | | | | | | | | |
|---|---|---|---|---|---|---|---|---|---|
| *Query = "cat"* | | | | | | | | | |
| Groundtruth | cat | feline | carnivore | placental | mammal | vertebrate | chordate | animal | organism |
| Regular sampling | cat | feline | carnivore | placental | mammal | wildcat | domestic cat | vertebrate | canine |
| Pretrain-finetune | cat | feline | chordate | vertebrate | animal | placental | mammal | carnivore | wildcat |
| *Query = "recliner"* | | | | | | | | | |
| Groundtruth | recliner | armchair | chair | seat | furniture | furnishing | instrumentality | artifact | whole |
| Regular sampling | recliner | armchair | seat | chair | furnishing | furniture | article | ware | toy dog |
| Pretrain-finetune | recliner | armchair | seat | chair | furnishing | furniture | instrumentality | artifact | cleaning pad |
| *Query = "motorist"* | | | | | | | | | |
| Groundtruth | motorist | driver | operator | causal agent | physical entity | | | | |
| Regular sampling | motorist | operator | driver | floridian | foe | | | | |
| Pretrain-finetune | motorist | operator | driver | physical entity | causal agent | | | | |

Where we used the bounds on $d$. Next, for case two, when $j \notin S_i$, using the events $\mathcal{E}_1, \mathcal{E}_2$, we have

$$\left| \left\langle \frac{q_i}{\|q_i\|_2}, \frac{x_j}{\|x_j\|_2} \right\rangle \right| < \frac{1}{\|q_i\|_2 \|x_j\|_2} |\langle q_i, x_j \rangle| \leq \frac{3}{2d} 100\sqrt{d \log n} = \frac{150}{\sqrt{C}\gamma} \leq r/2 \qquad (6)$$

Where we took $C > (2 \cdot 4 \cdot 150)^2$, which completes the proof as $r/2 < r - \epsilon$. Finally, note for runtime, one needs only generate $O(m)$ $d$-dimensional Gaussian vectors, and then compute each $q_i$ which takes $\tilde{O}(sd)$ time each, thus the total time is $\tilde{O}(md + nsd)$ as desired.

$\square$

## B   Implementation Details on ESCI Dataset

Here we provide additional details for experiments on ESCI. We use the SentencePiece model to tokenize the queries and products which are fed to standard 8-layer Transformers as the architecture for the encoder models in DE. For the Transformer, we use model dimension 512, 8 attention heads, two-layer MLP with GELU activation and a hidden dimension of 4096 as the feedforward network. The output embeddings from the Transformer are mean-pooled and projected to 128 dimensions, followed by a normalization to the unit $\ell_2$ sphere as the final embedding. The model is trained with the Lazy Adam optimizer [3] with a warmup stage of 2000 steps to a learning rate of 1e-4, followed by a linear decay to 1e-6 at step 50000.

## C   Additional Experiments for the Toy Setup in Section 4

In Figure 6, we provide additional results complementing Figure 3b. These results reconfirm that rebalanced sampling cannot effectively solve the lost-in-the-long-distance issue.

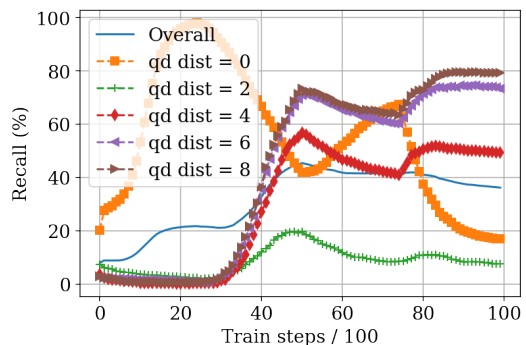

Figure 7: Retrieval quality at varying query-document distances on WordNet, **using hyperbolic embeddings of dimension** 16. The peak performance is a recall of 45.4% obtained at step 49. This recall is higher than that with the Euclidean embeddings of the same dimension *with standard training*, which is at 43.0% (see Table 1), demonstrating the superiority of hyperbolic spaces. Nonetheless, it is worse compared to our pretrain-finetune recipe, which has a recall of 60.1%.

## D   Comparison with Hyperbolic Embeddings on WordNet

For hierarchical relations, hyperbolic space is a popular choice for addressing the shortcomings of the Euclidean space [24]. Unfortunately, practical large-scale retrieval systems cannot widely adopt hyperbolic embeddings, due to a lack of an efficient approximate k-nearest neighbor search algorithms in hyperbolic spaces. Nonetheless, for the purpose of scientifically understanding the capability of hyperbolic geometry for solving the HR problem, here we implement hyperbolic embeddings and perform experiments on the WordNet dataset.

Specifically, we train 16-dimensional hyperbolic embeddings on the same 10M normal sampling data as in Section 6. Among many options for implementing hyperbolic embeddings, we use the reparameterization form in [12], using a learning rate of 0.01. All other training details are the same as those for Euclidean embeddings.

We evaluate the recall for query-document pairs at varying distances and report the results in Figure 7. We see that hyperbolic embeddings struggle to obtain a good balance between pairs with short vs long distances. Specifically, the model first learns to retrieve pairs with short distances, i.e. with distance 0 and 2. As it starts to retrieve longer distance pairs, the quality on the short distance pairs drops rapidly. The best overall recall (i.e., averaged over all distances) is 45.4% obtained at step 49. This recall is better than that of a DE trained on the same data, which is 43.0% (see Table 1), showing the superiority of embedding in hyperbolic space. However, it is still worse than our pretrain-finetune approach, which obtains a recall of 60.1%.

## E   Ablation Studies for Finetuning on WordNet

In this section, we study the effect of hyper-parameters in our pretrain-finetune recipe for the WordNet experiments presented in Section 6. In particular, the results in Table 1 are obtained with a finetuning learning rate that is 0.001 times the one used during pretraining, and a temperature that is increased from 20 during pretraining to 500 when finetuning. Here, we vary the choice of this learning rate multiplier and temperature during finetuning, and present the results in Figure 8.

For varying learning rate multiplier (see Figure 8a), we observe that the recall on long distance pairs improves as this multiplier is increased from a very small number of 1e-7 up to 1e-3. Crucially, we observe that the recall on short distance pairs are not significantly affected, despite the fact that such pairs are not included in the finetuning data. However, when this multiplier is further increased from 1e-3, the model performance starts to deteriorate on both short and long distance pairs.

In terms of temperature (see Figure 8b), we see that the best recall is obtained with a temperature of around 500. Both much smaller and much larger values of temperature lead to a quality loss.

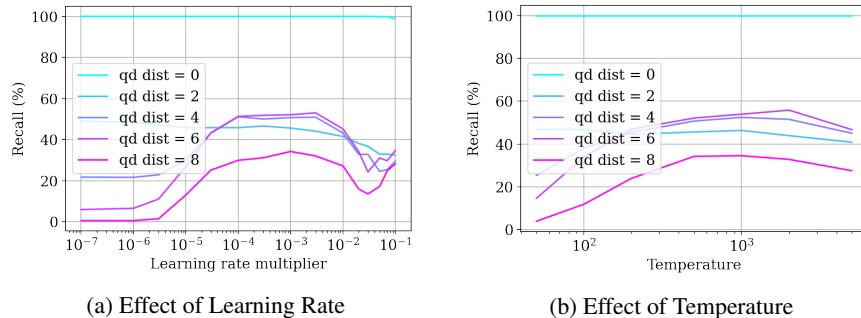

(a) Effect of Learning Rate

(b) Effect of Temperature

Figure 8: Effect of learning rate and temperature during the finetuning stage of the pretrain-finetune recipe for HR on WordNet.

Finally, Figure 8 shows that the model quality in terms of recall is not sensitive to the choice of these two hyper-parameters, making our method practically easy to tune.

