# OpenReview forum: "Hierarchical Retrieval: The Geometry and a Pretrain-Finetune Recipe"
_NeurIPS.cc/2025/Conference — NeurIPS 2025 poster_

### Official Review · Reviewer_zvXP · 2025-06-29

**Clarity:** 3
**Significance:** 2
**Originality:** 3
**Rating:** 4
**Confidence:** 4

**Summary:**

The paper considers hierarchical retrieval where query and documents are organized in a DAG. The paper proves that hierarchical retrieval can be learned with dual encoder through a constructive algorithm, then identified a lost-in-the-long-distance problem under a standard retrieval learning procedure. After that a pretrain-finetuning method is proposed to fix the problem. Two datasets are considered and goo results are shown. However, there are also some concerns on the proposed approach.

**Questions:**

1. How to apply the method without distances on the DAG as input?
2. Any intuition why the proposed pretraining-finetuning works better?
3. discuss the requirements for pretraining-finetuning

**Ethical Concerns:**

["NO or VERY MINOR ethics concerns only"]

**Final Justification:**

my concerns are addressed

**Limitations:**

limitation discussion is not complete, for example, one apparent limitation is that the proposed pretraining-finetuning method may not work well if the number of exact match data point is limited.

**Quality:**

2

**Strengths And Weaknesses:**

Strong points:
1. The paper is well organized and informative. While the proposed pretrain-finetuning procedure is simple, the process of developing it is informative and I learned something new from it.
2. The paper proves that dual encoders are feasible to solve hierarchical retrieval as long as the embedding dimension is large enough.
3. A simple pretrain-finetuning is proposed to fix the lost-in-the-long-distance problem and good empirical results are shown.

Weak points:
1. It is claimed that the algorithm does not require DAG as input, however, the main algorithm in Section 5.2 requires the distance on the DAG as input. In real-world applications, without the DAG, how can one get the distances?
2. It is unclear why pretrain-finetuning would fix the lost-in-the-long-distance problem? Why it is better than mixed training? Is there any intuition on this?
3. There are prerequisites for pretrain-finetuning to work better than mixed training, but it is not discussed at all in the paper. For example, one simple case where there are only a few data point for exact match, so that the model is under-trained when pre-training only on the exact match data, I would imaging pretrain-finetuning to be worse than mixed training. It would be better to discuss when pretrain-finetuning works better.

---

> ### Author Rebuttal · Authors · 2025-07-31
>
> We thank the reviewer for the thoughtful review and for finding our paper "well organized and informative”, the feasibility proof valuable, and the empirical results compelling.
>
> **How to Apply the Method without DAG Distances**
>
> This is an excellent point, and we apologize for the lack of clarity. The key point, which we will clarify in the revision, is that our recipe does not require the full DAG or precise path lengths. It only requires a proxy for distance that allows us to partition the training data into two sets: a "short-distance" (or "easy") set for pretraining, and a "long-distance" (or "hard") set for finetuning. This proxy can be readily available in practice or easy to obtain:
> On the ESCI shopping dataset, the dataset's labels directly provide this split: we treat "Exact" matches as short-distance and "Substitute" matches as long-distance.
> In some applications, one may be able to use a simple heuristic (e.g., direct parent-child links vs. all other ancestors) to create the finetuning set.
> Human annotation of the training dataset into short vs long distance pairs is another option and such a binary annotation is significantly easier to obtain than constructing the full DAG.
>
> **Intuition for Why Pretrain-Finetune Works Better**
>
> As our experiments show, mixed/rebalanced training forces the model to learn two different structures simultaneously—the local structure of exact matches and the global structure of distant ancestors. This creates a difficult optimization trade-off, where improving on one often comes at the expense of the other (as seen in Figure 3b and Table 1).
> Our two-stage recipe decouples these objectives. The pretraining stage grounds the model by learning robust embeddings for the local, short-distance structure. The finetuning stage then acts as a gentle adjustment, adapting these well-formed embeddings to the more challenging long-distance pairs without catastrophically forgetting the initial structure.
>
> **On Prerequisites and Limitations**
>
> This is a valid concern and an important limitation. Our recipe implicitly assumes that there is sufficient "short-distance" data to learn a meaningful initial representation during pretraining. If this data is extremely sparse, pretraining may be ineffective, and a mixed training approach might indeed perform better. We thank the reviewer for this insight. We will explicitly add this to our section 5.2, stating that the effectiveness of our recipe depends on having a reasonable amount of data for the initial pretraining stage.

---

> > ### Comment · Reviewer_zvXP · 2025-08-04
> >
> > I thank the authors for the response. My concerns are addressed and I will increase the rating.

---

### Official Review · Reviewer_hWtC · 2025-07-03

**Clarity:** 3
**Significance:** 3
**Originality:** 3
**Rating:** 4
**Confidence:** 3

**Summary:**

This paper studies hierarchical retrieval (HR), where documents are organized in a hierarchical structure and the goal is to retrieve all ancestors of a query in the hierarchy. The authors first prove that dual encoders are theoretically feasible for HR when the embedding dimension is linear in hierarchy depth and logarithmic in document count. They then identify a "lost-in-the-long-distance" phenomenon where retrieval accuracy degrades for documents further away in the hierarchy. To address this, they propose a pretrain-finetune recipe that first pretrains on regular data then finetunes on long-distance pairs. Experiments on demonstrate that this approach significantly improves long-distance retrieval performance.

**Questions:**

1. Generalization: How does the pretrain-finetune recipe perform on hierarchies with different structural properties (e.g., varying branching factors, irregular depth)? What are the key hierarchy characteristics that determine success?

2. Computational analysis: What is the computational overhead of the two-stage training compared to standard approaches? How does the method scale with hierarchy size and depth?

**Ethical Concerns:**

["NO or VERY MINOR ethics concerns only"]

**Limitations:**

Yes.

**Paper Formatting Concerns:**

The paper generally follows NeurIPS formatting guidelines.

**Quality:**

4

**Strengths And Weaknesses:**

Quality:

Strengths: Rigorous theoretical analysis with constructive proof establishing feasibility bounds; systematic experimental validation from synthetic to real data; comprehensive evaluation across multiple metrics and datasets

Weaknesses: No statistical significance/cost analysis

Clarity:

Strengths: Well-motivated problem with clear real-world applications; excellent progression from theory to synthetic experiments to real applications; clear mathematical exposition with intuitive explanations

Significance:

Strengths: Highlights a practical failure mode for DEs and offers a simple fix requiring no model changes.

Weaknesses: Limited scope of evaluation (only hierarchical retrieval tasks); unclear generalization to other asymmetric retrieval problems;  unclear impact for very large corpora or LLM-based encoders

Originality:

Strengths: First formal margin bound linking hierarchy depth to DE dimension

Weaknesses: Pretrain/finetune idea is incremental relative to curriculum or hard-negative mining

---

> ### Author Rebuttal · Authors · 2025-07-31
>
> We are very grateful to the reviewer for the thorough review and the positive assessment of our paper's quality, clarity, and originality. We appreciate that you found our theoretical analysis "rigorous" and our contribution of identifying a failure mode and offering a simple fix to be significant.
>
> **Generalization to Different Hierarchies**
>
> While our synthetic experiments use perfect trees for controlled analysis, our main real-world experiment is on WordNet, which is a large, complex, and irregular directed acyclic graph (DAG), not a simple tree. The fact that our method yields significant improvements on WordNet (e.g., boosting recall on long-distance pairs from 19% to 76%) demonstrates its effectiveness on real-world, non-uniform hierarchies.
>
> **Computational Overhead**
>
> At inference time, our method has zero overhead. The final model is a standard dual encoder, and its retrieval latency is identical to that of a model trained via a single-stage process. The overhead exists only during training, as our method involves two stages. On WordNet, pretraining takes 50k training steps while finetuning only takes 700. On ESCI, pretraining and finetuning takes 30k and 50k training steps, respectively.
>
> **Statistical Significance**
>
> We acknowledge in our paper checklist that we did not report error bars or formal statistical significance tests. We agree this would strengthen the paper. However, we would like to point out that the performance improvements shown are very large and consistent across multiple datasets and embedding dimensions. For example, in Table 1 (d=64), our method improves the minimum recall from 19.4% to 75.7% and the overall recall from 71.4% to 92.3%. We believe these substantial gaps provide strong evidence for the effectiveness of our approach.
>
> **Novelty of Pretrain-Finetune**
>
> While pretrain-finetune paradigm is common in ML, it is used primarily in the context of pretraining a large model followed by adapting such a model to a particular downstream task. It has never been used to address the lost-in-the-long-distance issue.
> We see our primary contributions not as the invention of this paradigm, but as: (1) discovering and articulating the "lost-in-the-long-distance" phenomenon as a key failure mode for DEs in hierarchical retrieval, and (2) identifying that this recipe offers a simple and effective solution, especially when contrasted with the more intuitive but much less effective rebalancing strategy.

---

> > ### Comment · Reviewer_hWtC · 2025-08-05
> >
> > Thanks for replying and I will keep my original recommendation.

---

### Official Review · Reviewer_XaUt · 2025-07-04

**Clarity:** 3
**Significance:** 3
**Originality:** 3
**Rating:** 4
**Confidence:** 3

**Summary:**

The paper studies a new model of "hierarchical retrieval" and dual encoder design and training for it. In normal retrieval, the dual encoder is trained with positive example {(q,d+)} to predict relevant docs for a query. This paper extends this to the case where queries form a somewhat hierarchical structure, or rather a DAG. If there is an edge from q1->q2 in the DAG, docs relevant for q1 must also be relevant to q2. The motivation seems reasonable in that we often want to select, and refine relevance matches.

The paper first studies an entirely toy dataset and simple tree-structured DAG, and studies simple random guassian and learn embeddings. The paper projects and matches the dimensionality needed to get 95% recall. The paper also studies the depth and imbalance of DAG in synthetic settings, and offers ideas on how to adjust the training. The paper then studies wordnet dataset and relations between word categories and trains a model that achieves reasonable accuracy for it.

What is not clear is how this applies to the e-commerce scenarios that motivate the problem in intro. If I had a shopping catalog with more complex documents with their own multi-modal embeddings, and weak signal, with multiple docs matched per query category, how does this pipeline extend and scale?

**Questions:**

-  Can this formulation extend to product catalogs and multi-modal docs and nebulous query intent. What if the DAG is over precisely stated query categories, but the queries are slightly variants of categories.
update: Authors state this is possible. Concrete proof might be beyond the scope of this paper.

- Are there alternate ways of training for this problem and how does this approach compare? For example, would treating DAG as flat structure and using extreme multi-label classification work?
update: Authors state that XMLC are impractical, however XMLC methods have been proved to be practical and are deployed at millisecond-latency at scale.

**Ethical Concerns:**

["NO or VERY MINOR ethics concerns only"]

**Final Justification:**

Updated the review. Disagree with on one of the rebuttal points, reflected that in update. Score unchanged.

**Quality:**

3

**Strengths And Weaknesses:**

Strengths:
- Interesting new variant of standard relevance problem
- good set of toy experiments that set up for the real problem
- A good answer for small real world datasets

Weakness
- Unclear whether this formulation can take on and scale to real-world problems
- Are there alternate ways of training for this problem and how does this approach compare?

---

> ### Author Rebuttal · Authors · 2025-07-31
>
> We thank the reviewer for the positive feedback and for finding our problem variant "interesting" and our experiments well-designed.
>
> **Extending to Complex, Real-World Scenarios**
>
> This is an excellent question that gets to the heart of practical applicability. Our approach is highly adaptable precisely because the dual encoder (DE) architecture is modular. The core of our contribution is the pretrain-finetune training recipe, which is agnostic to the specifics of the encoders. In our work, we use lookup tables for the WordNet experiments and modern Transformer-based text encoders for the ESCI shopping dataset. For a multi-modal product catalog, these encoders could be readily replaced with architectures that process both text and images (e.g., CLIP-style models) without changing the fundamental training strategy: pretrain on "exact" or "primary" matches, then finetune on "substitute," "broader," or other "long-distance" matches. The ESCI experiment is a direct step toward showing this real-world applicability.
>
> **Alternate Training Formulations**
>
> We thank the reviewer for this insightful question about alternate training formulations. It is indeed possible, as the reviewer suggests, to frame our Hierarchical Retrieval (HR) task as an Extreme Multi-Label Classification (XMLC) problem by "flattening" the document hierarchy and treating each document as an independent label.
> However, this approach could be suboptimal as many common XMLC methods are not well-suited for the practical constraints of large-scale retrieval systems, for reasons beyond just the loss of the hierarchical structure.
>
> - Modern retrieval systems require millisecond-level latency. Alternate XMLC methods, such as One-vs-All (OVA) makes a prediction by running the query through a unique classifier for every document in the corpus. The linear cost of this operation is too slow for any real-time application.
>
> - Moreover, real-world document collections are constantly growing. With our embedding-based approach, adding a new document is trivial: we generate its embedding and add it to the ANN index without retraining the model. In contrast, adding a new document (i.e., a new "label") to a tree-based or OVA-based XMLC model is far more complex and would likely require significant retraining or model restructuring.
>
> Therefore, while one could apply XMLC methods by flattening the hierarchy, this would not only discard the crucial hierarchical information that our method is designed to capture but would also likely lead to a system that is too slow and inflexible for a practical retrieval setting. We chose our approach because it both models the hierarchical relationships and aligns with the high-performance architecture of modern retrieval systems.

---

### Official Review · Reviewer_Dn6e · 2025-07-13

**Clarity:** 3
**Significance:** 2
**Originality:** 2
**Rating:** 4
**Confidence:** 3

**Summary:**

This paper studies dual encoder (DE) models for information retrieval in the specific context of Hierarchical Retrieval (HR), where documents are organized in a hidden hierarchy, and a query should retrieve all of its ancestor nodes. The authors first provide theoretical guarantees that DEs are capable of solving HR if the embedding dimension scales appropriately with the depth and size of the hierarchy. They then identify a practical challenge -- retrieving distant ancestors becomes harder (termed the "lost-in-the-long-distance" problem). To address this, they propose a pretrain-finetune strategy: first training on standard data and then fine-tuning on long-distance query-document pairs. Experiments on synthetic trees, WordNet, and ESCI demonstrate improved performance, particularly for long-distance retrieval. The work bridges theoretical insight with practical system challenges and offers an algorithmic tweak to improve DEs on hierarchically structured data.

**Questions:**

- Why is there no comparison with hyperbolic embedding models (e.g., Poincare and hyperbolic embeddings), which also aim to solve hierarchical representation issues? These seem like direct competitors.
- Can the authors elaborate on why the pretrain-finetune strategy works better than rebalancing? Is there an intuitive or theoretical justification?
- How sensitive is the performance to the specifics of the fine-tuning stage (e.g., temperature, learning rate, early stopping)? Could you add ablations?
- The theoretical results (Theorem 3.1) rely on access to S(q) sets. Can you discuss or experiment on how noisy estimation of these sets might affect the embedding quality?
- Could you provide retrieval time and latency comparisons with other methods, particularly for large datasets? This would help validate the practical relevance.

**Ethical Concerns:**

["NO or VERY MINOR ethics concerns only"]

**Final Justification:**

Resolved issues:
- Authors provided direct comparisons with hyperbolic embedding models
- Authors clarified the rationale behind the pretrain–finetune approach vs. rebalancing and provided sensitivity analyses.
- Authors explained that inference latency remains unchanged, preserving DE scalability.

Unresolved issues:
- The algorithmic novelty remains modest; the pretrain–finetune approach is conceptually simple and common in ML.
- Theorem 3.1 still relies on impractical assumptions (exact S(q) sets), limiting its direct applicability.
- Baseline coverage, while improved, could be broader and more systematically analyzed.

**Limitations:**

yes

**Paper Formatting Concerns:**

Null

**Quality:**

2

**Strengths And Weaknesses:**

Strengths:
- Provides a constructive theoretical proof showing DE feasibility for HR under certain embedding dimension assumptions.
- Empirical findings consistently validate the theoretical results across synthetic and real datasets.
- Identifies and articulates a realistic failure case (lost-in-the-long-distance) that affects DEs in HR tasks.
- Introduces a practical and simple pretrain-finetune training recipe to address the failure case.
- Experimental results show strong improvements in recall for distant matches, especially in the WordNet dataset.

Weaknesses:
- Lack of baselines: The paper omits comparison with hyperbolic or other hierarchical embedding models (e.g., Poincare embeddings), which are directly motivated by similar geometric considerations.
- The pretrain-finetune strategy is intuitive and potentially ad hoc. There is limited analysis on why it helps, beyond empirical results.
- The novelty of the proposed algorithmic contribution (pretraining then fine-tuning) is limited -- it is a common ML technique and might be seen as an incremental tweak.
- The theoretical analysis is insightful but assumes knowledge of the matching sets {S(q)}, which are not available in practical settings; this limits the practical relevance of Theorem 3.1.
- Some claims, like DEs outperforming handcrafted embeddings in low dimensions (Fig. 2), are underexplored and not connected back to theory.
- The ESCI experiment is a stretch from the core problem setting (HR), and the analogy to "long-distance" is loosely defined and not justified.

---

> ### Author Rebuttal · Authors · 2025-07-31
>
> We sincerely thank the reviewer for the detailed and insightful feedback. We appreciate the accurate summary of our work and the challenging questions, which will help us improve the paper significantly. We address the noted weaknesses and questions below.
>
> **Comparison with Hyperbolic Models**
>
> Hyperbolic models are not our direct competitors. While they are indeed well-suited for hierarchical data, as we note in Section 1.2, they are "NOT suited for retrieval due to a lack of an efficient nearest neighbor search that hinders their applicability to large scale data". Our goal was to investigate the capabilities and limitations of the widely-used Euclidean dual encoder (DE) architecture and provide a practical method to improve it without sacrificing its key advantage: scalability via efficient ANN search. We will revise Section 1.2 to make this motivation clearer and to better contextualize our work against these important baselines.
>
> That said, we are happy to include additional experimental results with Hyperbolic models to situate our method into a broader context for interested readers.
> - On HyperLex (see Table 2), Poincare embedding obtained a Spearman’s rho of 0.512 according to ref. [24], which is better than our method (with rho = 0.415).
> - On WordNet (see Table 1), we conducted an additional experiment with Poincare embedding using the reparameterization approach in ref. [12] for embedding dimension = 16. We obtained an overall recall of 45.4% which is better than standard training of DEs (43.0%), but is worse than our pretrain-finetune approach (60.1%). To obtain more insights into the behavior of Poincare embeddings, we plotted the recall at varying distances between query and document, and observed that the model struggles to obtain a good balance between pairs with short vs long distances. That is, the model first learns to retrieve pairs with short distances, i.e. with distance 0 and 2. As it starts to retrieve longer distance pairs, the quality of the short distance pairs drops.
> We will include these results in the revised version.
>
> **Novelty of the Pretrain-Finetune Recipe**
>
> While pretrain-finetune paradigm is common in ML, it is used primarily in the context of pretraining a large model followed by adapting such a model to a particular downstream task. It has never been used to address the lost-in-the-long-distance issue.
> We see our primary contributions not as the invention of this paradigm, but as: (1) discovering and articulating the "lost-in-the-long-distance" phenomenon as a key failure mode for DEs in hierarchical retrieval , and (2) identifying that this recipe offers a simple and effective solution, especially when contrasted with the more intuitive but much less effective rebalancing strategy.
>
> **Intuition on Pretrain-Finetune Recipe vs Rebalancing**
>
> Our experiments show that rebalancing improves long-distance recall at the severe cost of short-distance performance (Figure 3b , Table 1). We hypothesize this is because the model is forced to solve a difficult trade-off between two distinct objectives simultaneously. Our two-stage recipe decouples these objectives. Pretraining first grounds the embeddings in the local, short-distance structure. The finetuning stage then gently adjusts these embeddings to accommodate the global hierarchy, improving long-distance recall while preserving the already-learned local structure (Figure 3c).
>
> **Practicality of Theorem 3.1**
>
> The theorem's purpose is to provide a formal existence proof that answers our first key question, "Q1: Does there exist Dual Encoders that solve Hierarchical Retrieval?". By proving feasibility, it motivates the subsequent empirical investigation into whether such embeddings can be learned from data, which is the practical focus of Sections 4-6.
>
> **Retrieval time and latency**
>
> At inference time, the model is a standard DE. Therefore, there is no change in retrieval latency compared to a conventionally trained model, which is a key practical advantage. The computational difference is in the training, which is a two-stage process. We will clarify this in the paper.
>
> **Ablations**
>
> In response to this great comment, we conducted sensitivity analysis on the effect of learning rate and temperature during finetuning for our HR on WordNet experiments.
> - For varying learning rates, we observe that the averaged recall across all distances are within 1% difference for a wide range of learning rate choices in the range of 1e-4 to 3e-3 (main results in the paper are reported for 1e-3).
> - For varying temperatures, we observe that the averaged recall across all distances are within 1% difference for a wide range of temperature choices in the range of 500 to 2000 (main results in the paper are reported for 500).
>
> We will include this discussion and the corresponding plots in the revised version.

---

> > ### Author Response · Authors · 2025-08-05
> >
> > Dear Reviewer Dn6e,
> >
> > Thank you again for your time and for providing a detailed and insightful review of our paper.
> >
> > We are writing to you as the author-reviewer discussion period is ending soon. We have posted a response addressing the valuable points you and the other reviewers raised, and we wanted to follow up respectfully.
> >
> > In our response, we specifically addressed your questions regarding the comparison to hyperbolic models (discussing the trade-off with retrieval scalability), provided a deeper justification for our pretrain-finetune recipe over rebalancing techniques , and clarified the role of our theoretical results in establishing feasibility.
> >
> > We would be very grateful to know if our response has addressed your main concerns, or if you have any further questions.
> >
> > Sincerely,
> > The Authors

---

### Decision · Program_Chairs · 2025-09-17

**Decision:**

Accept (poster)

**Comment:**

The paper studies hierarchical retrieval (HR) with dual encoders (DEs). It (i) provides a constructive feasibility result relating HR solvability to embedding dimension (linear in depth, logarithmic in corpus size), (ii) empirically identifies a “lost-in-the-long-distance” failure mode where distant ancestors are under-retrieved, and (iii) proposes a simple pretrain-finetune recipe that substantially improves long-distance recall while preserving near-distance performance. Experiments span synthetic trees, WordNet, and an e-commerce (ESCI) setting.

Review synthesis:
- Positives (broadly shared): clear writing and problem framing; a useful theoretical existence result; well-designed empirical study that isolates a practical failure case; simple training recipe with strong gains on long-distance recall; retention of DE latency/scalability at inference.
- Concerns (initial): limited baselines (notably hyperbolic/hierarchical embeddings), modest algorithmic novelty (recipe might appear ad hoc), practicality of the theorem (assumes access to sets/distances), and questions about real-world scalability and alternative formalisms (e.g., XMLC).
- After rebuttal: Authors added comparisons to hyperbolic models, provided sensitivity analyses (LR/temperature), clarified that inference latency is unchanged, and articulated when/why the two-stage recipe can outperform rebalancing. Several reviewers increased or maintained borderline-accept scores; remaining concerns focus on modest novelty and the practicality limits of the theorem.


Remaining weaknesses:
- Novelty: The training recipe is incremental conceptually, though well-motivated and effective.
- Theory–practice gap: The feasibility theorem relies on information (exact sets/distances) that may be unavailable; its role is primarily existential.
- Scope: Results are concentrated on HR; broader generalization (to other asymmetric retrieval tasks, very large corpora, and multi-modal encoders) is argued but not fully demonstrated.
- Alternatives: XMLC and other pipelines are not systematically benchmarked at scale; one reviewer notes their practicality at ms-latency.

**Camera-ready suggestions**:

Integrate the new hyperbolic baselines and distance-stratified plots; clearly position them w.r.t. DE scalability and ANN search.

Expand sensitivity/stability analyses (learning rate, temperature, fine-tuning steps) and report statistical significance/error bars where feasible.

Make explicit the data prerequisites (how to obtain short/long partitions; failure modes when “short-distance” data is scarce).

Clarify the theorem’s role (existential) and discuss how approximate distance proxies affect learning.

Add a concise discussion/experiment contrasting against a representative XMLC baseline at realistic scale/latency, or justify its omission with concrete system constraints.